# A Qualitative Study of Adolescents from Refugee Backgrounds Living in Australia: Identity and Resettlement

**DOI:** 10.3390/ijerph21030281

**Published:** 2024-02-28

**Authors:** Nigar G. Khawaja, Robert D. Schweitzer

**Affiliations:** School of Psychology and Counselling, Queensland University of Technology, Victoria Park Road, Kelvin Grove, QLD 4059, Australia; r.schweitzer@qut.edu.au

**Keywords:** acculturation, adolescent, Australian, identity, meaning-making, refugee, resettlement

## Abstract

Identity formation among young people from refugee backgrounds is complex, as it occurs while they are both integrating into a novel cultural landscape and navigating the intricacies of adolescence. The present study explored sense of identity and experiences among refugee youth in the context of resettlement. Nineteen young people (15–18 years) from refugee backgrounds, settled in Brisbane, Australia, took part in the study. An individual semi-structured interview, drawing upon the Tree of Life method, was used. The thematic analysis (TA) methodology was adopted, and several themes emerged: experiencing changes in family roles; experience of belonging; experience of bonds with lost loved ones; dealing with emotions in a new context; experience of self in the context of change. There was one emergent overarching theme of meaning-making in the context of change. These themes were explicated within the framework of social identity and sociocultural theories, which emphasises the dynamic co-construction of identity through the interplay of belonging and meaning-making within specific contextual settings. This study highlights the fundamental role of social context, particularly the fostering of school belonging, in the multifaceted process of identity construction. The findings identify the importance of integrating multiple identities and experiences to develop a comprehensive and resilient sense of personal cohesion and selfhood.

## 1. Introduction

Identity formation is an intricate process for all adolescents [1]. In the context of young people from refugee backgrounds, an additional layer of complexity emerges, given the interplay of various psychological, social, cultural, ethnic, religious, and personal factors which contribute to identity [2]. While the field of identity research is heterogenous, there is a consensus that life experience plays a central role in the formation of identity [1,3]. For adolescents from refugee backgrounds, rupturing life experiences are recurrent and profound. Trauma, loss, displacement, and resettlement all have the potential to cause immense disruption to young people’s sense of personal coherence, bringing identity issues to the fore [2]. In addition, the increased salience of cultural context and unfamiliar group membership in the position of other [4], as well as pre-migration experiences, acculturation processes, and post-settlement challenges exert significant influences on the complicated tapestry of identity development [2].

This paper seeks to explore and articulate the experiences of identity and resettlement among refugee youth. It achieves this by identifying and interpreting qualitative data obtained from recently resettled young individuals from refugee backgrounds in Brisbane, Australia. The data analysis has been informed by social identity and sociocultural theories that amalgamate insights from anthropology, sociology, and psychology to the understanding of the construction of self and identity. Such theories emphasise the notion of co-construction in identity formation, that is, the importance of the dynamic relationship between an individual and their environment in shaping their sense of self [5]. In essence, co-construction of identity is not an individual effort but involves the reciprocal influence of collective and personal narratives, values, beliefs, and experiences which are internalised and adapted [6]. Hence, identity formation for multicultural individuals is influenced by both the persons’ new cultural context and their “home” culture. Social identity theory (SIT), a widely known theoretical framework, lends itself to an understanding of how individuals from refugee backgrounds may renegotiate their social identities within their host countries. Further, identity is understood as being constructed in sociocultural settings.

### 1.1. Theoretical Perspectives

#### 1.1.1. Social Identity Theory

Social identity theory (SIT) is a widely acclaimed theory; according to SIT, individuals ascertain part of their identity from their perceived affiliation with relevant social groups [7]. As a part of growing up, young persons are part of family, school, and several interest, leisure, religious, ethnic, and community groups. They interact with these groups and belong to these groups at various points in their lives [8]. These social interactions can be positive or negative. The positive interactions can lead to a tendency for people to express a preference for in-group members compared to out-group members [9]. Along with personal identity and achievements, the group affiliation and the achievements of the group reinforce a positive social identity and foster self-esteem [10]. Group memberships are associated with a sense of belonging and connectedness [3]. Conversely, if a negative social identity is maintained through experience, people may disassociate from their group. Thus, social identity is subject to fluctuations based on experience and meaning-making. It is further elaborated upon from the sociocultural perspective.

#### 1.1.2. The Sociocultural Perspective

Within this model, the sense of self and environment are intrinsically linked and interdependent, with human action shaped by sociocultural context and vice versa [11]. Individuals exist in “figured worlds”, that is, the socially and culturally constructed lenses through which we see and engage with the world [12]. In this framework, particular aspects of experience are assigned significance and value [12]. By thinking, feeling, and behaving in line with a particular template or set of rules, the figured world is preserved and further developed [13]. When individual and collective interpretations within our figured world conflict with one another, we develop new ways of thinking, feeling, and behaving, which in turn, constitute identity [14]. Consequently, we are constantly renegotiating our identity to accommodate new experiences and the meanings we derive from these experiences [15].

Identity formation is the never-ending process of integration and reconstruction of our own unique life experiences [15]. When we engage in this process, our own subjective meaning of the experience is removed from the context in which it emerged and “placed in meaningful relation … to the existing semiotic architecture of the self in generalized form” [16]. Thus, we are constantly engaging with the world, in a perpetual state of flux, and reconstructing our self-structure to accommodate the subjective meanings derived from each new experience [17,18]. Generally, this process is not conscious. However, under certain circumstances, this process becomes overt and more deliberate, as we try to derive meaning from experiences that do not fit neatly into our carefully constructed mosaic, comprising a number of self-states.

#### 1.1.3. Identity of Young People from Refugee Backgrounds

Young people from refugee backgrounds are members of multiple groups. They have families, which could be atypical because of the traumatic loss of family members [19]. Despite varying life circumstances, they are all part of educational institutions. Further, they tend to have affiliations with ethnic and cultural groups [20]. Their identity is considered to develop through their participation with their cultural traditions, norms, roles, and contexts [21]. Further, this social participation exposes them to cultural values and beliefs which influence the way they give meaning to their own self and construct their identities. Identity is shaped by historical circumstances [22]. Young people with refugee backgrounds may be directly or indirectly impacted by war, trauma, and other atrocities. These young people were originally members of collectivistic, authoritarian, and hierarchical societies [6]. Therefore, they may see themselves as a part of their heritage group. They may look up to authority figures for support and direction, and may accept specific roles in hierarchical setups that are geared towards caring for their families and communities [23].

As the social environment changes in one’s life, one’s identity is also subject to change [1,6]. As young people from refugee backgrounds enter a highly developed and individualistic Australian society, their identity may undergo a shift. As they interact with a different language, culture, religious beliefs, and norms, they have to navigate their new environment by constructing a hybrid identity [10]. The identification of their personal, ethnic, religious, and national context may have to be negotiated [2,15]. This shift in sociocultural context and group loyalties is further impacted by potential trauma and adverse experiences, influencing social networks and interactions within communities [21]. Under these circumstances, young people from refugee background may encounter “rupturing otherness” [24]. However, these rupturing experiences are managed to regain a sense of coherence. For young people who are refugees in particular, these repeated adversities are likely to profoundly disrupt their self-structure. Consequently, it is assumed that the process of identity construction will frequently enter their conscious awareness [4,25]. Refugees who successfully integrate their experiences and context tend to cultivate a more cohesive self-concept. This process has been described by Papadopoulos as adversity-activated development [26].

### 1.2. The Current Study

Presently, there is a paucity of research looking specifically at young people from refugee backgrounds, particularly in Australia. The current study was designed to be exploratory in nature, adopting an inductive approach, placing the voices of the participants at the forefront. The study involved guided discussion around how young people from refugee backgrounds see themselves and perceive the world around them, thus reflecting their sense of self or identity in the context of resettlement. By explicating themes related to identity and resettlement, the current study contributes to the existing body of literature on identity among refugee youth and identifies issues that may be unique to participants in an Australian context.

## 2. Method

### 2.1. Design

The current study utilised a qualitative approach with a semi-structured interview based on the narrative Tree of Life tool [27,28]. This qualitative research methodology was selected for the following reasons: (1) to enhance participant control over the research process, reducing the likelihood of a power imbalance occurring between participant and researcher [29]; (2) to allow for new information to be introduced by the young people [30].

### 2.2. Participants

Nineteen young people (13 females and 6 males) from refugee backgrounds, aged 15–18, participated in the study. The participants had migrated from 12 different countries across South Asia, Africa, and the Middle East. They had entered on humanitarian visas, with the majority of participants being granted a permanent visa. Table 1 provides an overview of participant demographic information. Pseudonyms are used to preserve their anonymity.

### 2.3. Materials

#### Tree of Life (ToL)

The interviews with the participating young people were structured around the Tree of Life tool (ToL) [27,28]. The template is based on guidelines provided by Ncube [28]. These guidelines were condensed into a template. Initially developed to assist children affected by HIV/AIDS in Southern Africa, the ToL is a narrative tool designed to facilitate self-awareness and create a more positive and coherent self-narrative [27,28,31].

The ToL uses the metaphor of a tree to map aspects of a young person’s life. In the ToL, the roots signify a person’s family, community, and/or place of origin, while the ground is representative of their current day-to-day routine (e.g., school, work, living arrangements). At the trunk of the ToL, individuals are invited to reflect on their strengths, which may include characteristics or personality traits that are seen as positive by the person (e.g., nurturing) or a particular skill (e.g., cooking). At the branches of the ToL, individuals are encouraged to reflect on their hopes and dreams for the future, while the leaves are reserved for people with whom the individual shares a significant relationship. Gifts received by others are represented in the fruit of the ToL. These include tangible gifts, as well as the fulfilment of emotional and interpersonal needs (e.g., love, support) and the modelling or transmission of certain values within a family or community. The ToL process provided the basis for the qualitative data underpinning the study.

Art Supplies: To complete the drawing component of the Tree of Life interview, all young people were provided with paper, pencils, an eraser, a pencil sharpener, and connector pens.

### 2.4. Procedure

#### 2.4.1. Ethics

Ethical approval was gained by the university’s research committee prior to the commencement of the study. Permission was also granted by the school at which the research was conducted.

#### 2.4.2. Recruitment

Participants were recruited from a transitional school for newly arrived refugee and migrant young people aged 11–18 years. The school was established to teach young people English, and to provide information and support to prepare them for life and study in Australia. Young people typically attend the school for a period of 6–18 months. Initial contact was made with teachers via email, with the facilitation of the school guidance officer. Relevant teachers and the school guidance officer were provided with information about what participation would involve and were given the opportunity to provide feedback on the research methodology to ensure that data collection methods were appropriate for the target population. Recruitment was undertaken over an extended period with sensitivity to the need to build rapport with potential participants prior to the commencement of research. That is, members of the research team had ongoing contact with the school for about five years and attended school functions and developed familiarity with members of the school community. This process was deemed essential in developing trust within the school community.

#### 2.4.3. Inclusion Criteria

Students were informed that they could participate in the study if they met the following criteria: (1) they were from a refugee background; (2) they were aged 15–18 years inclusive; (3) they were confident in their English language ability; (4) they had arrived in Australia on one of the humanitarian visas. Young people on work, migration, research, and training visas were excluded from the study. English language ability was desired to limit the dependence on the interpreter, as changes to language inevitably occur through the process of translation [32,33], which can alter the meaning of words or sentences [26]. All participants were recruited from advanced classes, which are reserved for students who have achieved the competencies required for the entry classes, ensuring intermediate or advanced English language ability. Participation was voluntary and no compensation was offered.

#### 2.4.4. Data Collection

Informed consent was obtained from the young people and their parents or guardians if necessary. The interviews were conducted by a postgraduate student completing her Professional Doctorate in Clinical Psychology. She had extensive experience working with vulnerable people with mental health issues and was supervised by the authors. She met with the participants individually and provided a summary of the study and information about confidentiality. The interviewer then met with the young people at the designated times and conducted semi-structured interviews and the drawing activity based on the ToL tool. The participants received instructions to draw a tree; it was explained that different parts of the tree tell us about different parts of ourselves. They were informed that the roots of the tree were about where they came from. The ground was explained as where they were lived currently, with whom they lived and what they did every day. As they moved up the tree, they had different parts to fill in. They were asked to draw and comment on what they were good at, different people in their lives, and the things these people have given them. It was explained that these things could be real things, like a bicycle, or it could be about something they have inherited, like their parents’ sense of humour, love, or support. The participants were informed that they could do the activity any way they wanted as there was no right or wrong way. The participants had the option of drawing or writing, and they were assured that the drawing or writing did not have to be perfect. The interviewer emphasised that it was important to learn about where the participant came from and what life was like for them at that point in time.

The interviewer asked the participant to elaborate on their drawing throughout the process (e.g., “Can you tell me more about where you come from?”) to promote and enhance discussion around identity and resettlement. Drawings were used to enhance the comfort of the participants and to use them as anchor points to have a conversation with the participant. Drawings and the associated verbal or written comments were interpreted along with the interview data. To facilitate communication, all participants were offered the option of using a phone interpreter; however, only one participant took up this option.

Interviews ranged from 16 to 82 min, with 18 of the 19 interviews conducted in English. One interview was conducted in Farsi and translated by a telephone interpreter service following a request by the participant. All interviews were audio-recorded and transcribed verbatim. The number of interviews completed was determined by the point at which data saturation occurred. To identify the saturation point, the researchers analysed and interpreted interview content throughout the data collection process, coding themes as they emerged. Once the researchers were satisfied that no new themes were emerging from the dataset (i.e., when content had reached the point of saturation), data collection was concluded.

### 2.5. Data Explication

The postgraduate student analysed the data in collaboration with the authors. Common themes related to identity and resettlement were explicated through a five-stage process of thematic analysis based on Braun and Clarke’s [34] method. Thematic analysis was selected as it is theoretically flexible and allows the researcher to code broadly and diversely using an inductive approach emerging from the data [34]. The process involved coding and collapsing the data before examining the more prominent themes in the context of the aims of the study, that is, to explicate themes related to identity and resettlement. This process was completed with the aid of a qualitative software analysis program, NVivo 12. In Stage 1, the student and the authors familiarised themselves with the data, listening to the recorded interviews and reading through interview transcripts whilst noting pertinent observations. In Stage 2, the student and the authors generated codes for the data, focusing primarily on features related to identity and resettlement, noting those that occurred repeatedly across the dataset. In Stage 3, themes were explicated, with existing codes collated to correspond with each theme. These themes were reviewed in Stage 4, with the authors examining each theme to determine whether they told a coherent story both independently and together. In Stage 5, the student and the authors defined and named each theme, focusing on the ‘essence’ of each theme and how they fit into the overall ‘story’ of the wider dataset. Due to the subjective nature of thematic analysis, it was anticipated that the personal biases, values, and judgments of the authors may significantly impact the coding and interpretation of data. To acknowledge these biases and how they shaped the data, analyses were documented in a reflexivity journal. The postgraduate student, who led this stage of the explication of the data, consulted regularly with the other authors to examine the data and discuss the themes, to further enhance the trustworthiness of the findings.

*Reflexive Comment:* The authors are academic researchers with a long-standing relationship with the school involved in the research. The first named author is female and was born in Pakistan and completed her post graduate training in clinical psychology in Australia, where she holds a senior position as an academic and research supervisor. She has lived experience of conflict and resettlement. The second author is a male academic, with a long history of researching refugee-related issues. The authors conceptualised the study and supervised the data collection and analyses. Both authors shared a commitment to undertaking research informed by principles of social justice and leading to practical outcomes.

## 3. Findings

### 3.1. Overview of Themes

Five prominent themes related to identity and resettlement were explicated: experiencing changes in family roles; experience of belonging; experience of bonds with lost loved ones; dealing with emotions in a new context; and experience of self in the context of change. These all fell within an overarching theme of meaning-making in the context of change. These themes are described below.

### 3.2. Experiencing Changes in Family Roles

Young people often spoke of role changes and shifts in the formation of their families. Due to the death or disappearance of the parents, half of the participants reported having an aunt or an uncle or an older sibling as a carer and guardian. This was a shift from the previous family structure and involved accepting a new person as a head of the family. For the majority of the participants, schooling had been disrupted in their home countries or previous countries of asylum. They reported not attending school at all or attending for some short periods of time. Subsequently, they were engaged in caring roles or domestic chores. Other members of their family had become accustomed to seeing them contributing to everyday responsibilities. According to one participant, “Back in Liberia… from school, I come home, go to market, buy food, cook, finish, that’s it” (Julliet). After settling in Australia, all the participants had to re-adjust to the role of being a full-time student. For some participants, the commitment to education made it impossible to fulfil roles they held previously in their country of origin, which they experienced as causing disruption to the family system. There was an expectation form the relatives that they would still complete their other duties. Juliet perceived her role in the family as disrupted by her return to full-time study, while Juliet’s uncle and aunt (carers) reportedly struggling to adjust to the loss of Juliet as caretaker. She reported:

Sometimes [my uncle] gets upset with me and then not talk to me … Like, when we were in Africa, [I was] working, washing … doing the housework … Not like that anymore … [Now] I have to go to school (Juliet).

Similarly, nearly half of the participants found that language barriers often precipitated a shift in family roles, with younger family members needing to interpret for the rest of the family:

Because [my elder sister/guardian] can’t speak English … when she has a problem, she talks to me and if there is no solution, I have to talk to another people, then they give me some idea, then I would just come and talk to her like my own idea … like when I first came to this school for interview and then they tried to ask her and she can’t speak the language and I talked to the teachers so they could understand (Lucy).

In some cases, the demand of language brokering kept them away from their education. According to Kalan, “I have to leave school for my father, for my mother, because they do not know the language”.

### 3.3. Experience of Belonging

A lived sense of belonging and being part of a group were common threads woven throughout participants’ narratives. Many of the young people described being displaced more than once and were often subjected to discrimination and racial abuse in previous countries of asylum. These experiences of displacement and discrimination often triggered a feeling of otherness, with the young people becoming acutely aware of issues surrounding their sense of difference and cultural identity.

When the participants arrived in Australia, the majority of the young people alluded to a strong sense of belonging within their school context, which was a transitional language school (whose population comprises refugees and migrants from different backgrounds). At school, the young people revealed a sense of unity in relation to shared identities of (a) migrant and (b) student.

Hani spoke very positively about her experiences of resettling in Australia (‘Everything is great … in Australia’) and did not share any difficulties associated with the transition. Another participant saw school as a place of support and comfort. According to her, “When I come to school, having my friends, laughing, and then I talk to the teachers sometimes and my teacher will have fun, laugh together” (Julliet).

### 3.4. Experience of Bonds with Lost Loved Ones

All participants reported having lost a significant relative or community member to war, disease, or physical separation. The lost family members were parents, grandparents or siblings (“Taliban killed my brother”—Kalan). For many of the young people, it felt important to sustain a bond with those who had passed away or been left behind. For some young people, the experience of bonds (or attachment) was about pursuing a dream shared with their lost loved one: “My elder brother … he liked [dentistry]. And he says for me, I have to become a dentist” (Nazir). For others, emotional bonds were experienced by internalising the traits or beliefs of the lost loved one. For example, many reported having adopted values of compassion and kindness, which they attributed to the influence of a lost parent, sibling, or other relative.

The loss of significant others was often associated with valuing a sense of responsibility, ambition, and diligence. The young people described feeling close to their lost loved one by acting in accordance with these values, thereby keeping the memory of the loved one alive in a country far from home. For many participants, carrying a loved one with them appeared to add value and meaning to their existence. Very often, the young people were able to intentionally integrate an aspect of their lost loved one into their identity without subjugating their own emerging values and beliefs. This was revealed by Kalan’s comment, who said “My brother said never go with the bad boys., You have to be a good man”.

### 3.5. Dealing with Emotions in a New Context

All the young people interviewed reported experiencing separation from friends and family in their home country or first country of asylum. A number of them talked about experiences of separation and how they considered themselves to be the lucky few who were able to relocate to a safer country. This ‘lucky one’ identity was linked with participants downplaying and potentially disavowing distressing emotional experiences and difficulties associated with resettlement. This phenomenon was particularly striking in young people whose futures appeared uncertain (e.g., those on temporary visas). Khin, for example, stated, “I am so happy in Australia”, followed by:

I’m so lucky to get here. Because you are now Australian, you’re freedom now, they tell me. They told me that. I can look my family. Because they are still in situation they are in now. They are still in refugee camp. How can we live? … Our people [are still in the] refugee camp … They tell me, you apply for us. I say I don’t have any visa now—you must wait 6 years … until I get citizenship … It’s good [to have that responsibility], it’s really good. Because I want that. My family is great (Khin).

During this conversation, Khin was noticeably distressed. However, he attributed his emotional ambivalence solely to the plight of the Rohingyan people and his family’s suffering in the refugee camp. Absent from Khin’s discussion were any distressing feelings attributable to resettlement in Australia, including the uncertainty associated with a bridging visa and the possibility of returning to the country of origin. This showed his capacity to disengage from potentially distressing experiences.

Further, Salim was on a Bridging Visa and his future was uncertain. His drawing (Figure 1) with black, white with grey shades, sharp lines and a stump near the tree, indicated underlying anxiety. Despite this uncertainty, he showed resilience by keeping himself in the present moment and focussing on his experiences in Australia. He reported, “I don’t think about the future. I just think about today and tomorrow. Nothing more than that”.

### 3.6. Experience of Self in the Context of Change

The process of meaning-making was implicit in the narratives of all participants. All participants shared a number of significant life events, including recurrent trauma, loss, displacement, and resettlement in Australia. In the context of identity, these events all have the potential to profoundly disrupt the person’s sense of personal coherence. So, what determines the likelihood of a rupture occurring? In the current study, the degree of disruption depended on the subjective meaning derived from the event. For example, Mairi, who had been in Australia for less than 12 months, stated:

In Iran, in the school, I’m best in class … but in Australia … I’m last one … so I’m very sad for this. Sometimes [I] cry, I can’t do it, anything in class and [my teacher] say [to] me, I’m listening … but … I can’t understand. I don’t like. I’m very sad … [I say to myself] why you can’t understand? … I don’t know why I [cannot understand] (Mairi).

In this excerpt, Mairi is struggling in an area of her identity, her capacity to learn, that had previously been very easy for her and evidently a great source of pride. The disparity between past self and current self can be seen as causing a rupture in Mairi’s self-structure, with Mairi unable to make sense of the disparity. This discontinuity in her sense of personal coherence leads her to feel emotionally distressed. Her drawing of the tree (Figure 2) also shows her frustration with acquiring English language and the impact it had on her sense of self.

Mairi’s experience stands in contrast to that of 15-year-old Kalan, who lost his elder brother in Afghanistan. Ordinarily, we would expect this loss to cause a rupture in personal coherence of some degree. However, Kalan recognised that his brother died because they lived in dangerous country:

My big mistake of my father is when he leave Pakistan [to go to Afghanistan] with … my big brother. Our life is good in [Pakistan] but … my father love his country [Afghanistan] too much (Kalan).

Kalan had already experienced significant trauma in Afghanistan (“They put bombs in our home”) and wanted to move abroad to safety. The meaning Kalan derived from his brother’s death aligned with his preexisting experience of Afghanistan, which prevented him from experiencing a profound rupture to his own sense of personal coherence. After fleeing Afghanistan, Kalan sustained a continuing bond with his deceased brother by striving for a more meaningful and successful life in a new country. Kalan made sense of his experience by continuing to carry his brother with him, which also assisted in the maintenance of a more cohesive identity, often referred to as a cohesive sense of self.

## 4. Discussion

While identity has been studied in several papers, the topic of identity in young people from refugee backgrounds has received less scrutiny. The current findings, drawing upon the explication of interview data, provide a novel exploration of the complex interplay between identity formation, meaning-making, and the resettlement experiences of young people from refugee backgrounds newly settled in Australia. The data highlight issues faced by young people during the early stages of relocating to a new country and the ways in which people construct meaning in the context of their reconstructed lives.

From a theoretical lens, the themes elucidated in the paper—addressing encompassing experiences of changes in family roles, experiences of belonging, experiences of bonds with lost loved ones, the process of dealing with emotions in a new context, and experience of self in the context of change—all contribute to the formation of a person’s identity, in terms of their membership of social groups [1,2,35]. Aligned with social identity theory, the interview data revealed that the participants adopted one or more social identities, mirroring their experience of repeated displacement and resettlement in different countries. Many of the young people also described their sense of self in dynamic terms, that is, as a process of change. This process is conceptualised within an overarching theme of ongoing meaning-making, as per social identity theory and sociocultural perspectives [15].

In Australia, young people from refugee backgrounds continue to occupy a position of marginalisation and difference; however, most participants in the current study did not refer to experiences of exclusion. In the current study, a sense of belonging strongly supported by the school appeared to buffer against feelings of otherness in the wider social context, with young people sharing at least two social identities of (a) student and (b) migrant. For young people from refugee backgrounds, school is one of the most important social contexts for promoting wellbeing and resettlement in a new country [36], with higher levels of connection and commitment to school reportedly being associated with lower levels of depression and higher levels of self-efficacy in young people with refugee backgrounds [37]. Unsurprisingly, school belonging has been found to weaken when transitioning from a specialist English Language School (ELS) to a mainstream school, with refugee young people in Australia reporting lower levels of support and higher levels of discrimination at mainstream schools compared to ELS schools [36]. Current findings lend support to the idea that feelings of marginalisation and difference might be attenuated by a strong sense of belonging within school and peer groups. However, more research into school belonging and sense of otherness is needed before firm conclusions can be drawn.

In the current study, meaning-making was implicit in all the young people’s narratives [17,18]. Across the dataset, participants shared several significant life events (e.g., loss, displacement, and resettlement). All these events had the potential to create a rupture in the person’s sense of coherence and sense of belonging. The extent of this rupture—if any—depended on the meaning derived from the experience. Consistent with the sociocultural context of identity theory and research, the young people in this study negotiated the restructure in the family and accepted people other than their parents as their carers and guardians. They were also motivated to make sense of their experiences, drawing on previously internalised voices and dialogues to determine how their experiences fit with their bricolage of self [15,38]. Young people may construct their sense of self by assembling different elements of their experience from their environment, sociocultural context, relationships, and personal factors, thereby creating a unique and multifaceted sense of identity [14,15].

With reported experiences of loss, continuing bonds with the loss of a loved one was a way of making sense of what had happened, with many young people internalising the beliefs and values held by lost loved ones. This finding is consistent with the contemporary bereavement literature [39], which suggests continuing bonds assist individuals to integrate and accept the loss of loved ones, so long as they accept that they cannot regain physical proximity with the person who is deceased.

Overall, participants shared experiences of emotional distress associated with rupturing life experiences that could not be placed meaningfully into the self-structure (e.g., the case of Mairi and Salim) [24]. For some young people, there was a tendency to engage in behaviour which may be understood in terms of disavowal [40], that is, to separate themselves from distressing emotions and/or difficulties associated with resettlement, as these did not fit with their self-identity of the ‘lucky ones’ who escaped war and persecution in their countries of origin.

The current findings inform our understanding of identity in a group of young people from refugee backgrounds. They support the idea that identity is constructed through a continuous process of configuration and re-configuration of self, usually in interaction with others, including their sociocultural context and where the person “fits” in relation to others. Additionally, the narratives of the participants illustrate a continual process of identity development, commencing in early childhood and supported by their social context (i.e., a specialist school dedicated to supporting youth with refugee backgrounds, newly arrived in Australia). The ultimate attainment of a fixed sense of identity endpoint remains uncertain; for these individuals, the evolution of their self-structure occurred in response to new experiences, alongside the development of collective memories. The group’s shared narrative implies an indefinite continuation of self-reconfiguration [41], supporting the idea that identity formation is an ongoing process and not a fixed entity [1,2,3,35].

From a clinical standpoint, these observations underscore the integral role of meaning-making in the resettlement and acculturation process. The findings highlight the need for clinical interventions that prioritise the exploration and interpretation of life experiences, supporting the construction a unified self-narrative. Moreover, providing education that underscores the intricate and multifaceted nature of identity and emotions could prove beneficial. This is particularly the case as the refugee young people in this study experienced distress, often associated with a sense of guilt, as they grappled with thoughts of those left behind in their countries of origin.

There were several challenges associated with research involving refugee young people. The current study took great care to manage these challenges throughout the research process, to improve the overall trustworthiness of the data and reduce the risk of research-related harm. To enhance the credibility of the data, the interviewer kept a reflexive journal and met regularly with the authors to discuss the interviews and the data [42]. To increase data integrity, the young people were reminded that participation was voluntary, and data would be de-identified. Participants were offered a range of response options (i.e., interview, drawing, writing) to enhance research control and autonomy, increasing the likelihood of participants feeling comfortable enough to provide rich and truthful information [36]. The semi-structured interview approach and ToL were both designed to be participant-centred, enhancing young people’s control over research participation, thereby shifting the balance of power between researcher and participant [29]. The inclusion of the ToL metaphor and the provision of the opportunity to draw or write allowed some emotional distance from more challenging topics [27,43], again increasing the likelihood of young people feeling safe enough to provide a rich and accurate picture.

### 4.1. Implications

The findings of this study have theoretical and practical implications. Firstly, the study highlights the importance of understanding changes in family roles. It is crucial to recognise the substantial role transitions that young people experience within their families, including the restructuring of the family as well as the shift from acting as caretakers and helpers in the household to becoming full-time students. The potential impact of these changes contributes not only to the individual’s social identities, but also the broader sociocultural construction of identity within the family unit. This is a valuable insight into the acculturation process and identity formation for youth with refugee backgrounds. Thus, it is important to provide support and resources to help young people navigate their new roles and responsibilities, balancing their familial and educational obligations within their cultural context.

The study endorses the significance of a sense of belongingness and connection for young people from refugee backgrounds. This is particularly relevant when considering their experiences of displacement and discrimination, and the subsequent possibility of a negative social identity. Experiences of belonging positively contribute to the health and wellbeing of refugee youth and facilitate the cultivation of a cohesive identity [3,22]. The availability of specialised schools in Australia plays a crucial role in this process. Hence, it is fundamental to create inclusive environments where young people feel welcomed, valued, and supported in exploring their cultural identities. Relatedly, the data promoted school connectedness. The school environment is seen as important in providing a supportive and inclusive environment for young people, where they can both transition into their new cultural milieu and develop social connections, academic skills, and a sense of purpose. A number of strategies may be implemented at a local level to enhance school belongingness and peer support, such as peer mentoring programs and cultural exchange activities within the school setting.

Lastly, the findings recognised the significance of the emotional bonds of young people from refugee backgrounds with lost loved ones. Importantly, they highlighted the deliberate integration of an aspect of the departed loved one into their identity. This result provides valuable insights into identity reconstruction in the context of loss. It is essential to provide opportunities for young people to respect and/or preserve the memories of their lost loved ones through activities such as storytelling, creative expression, or commemorative activities.

### 4.2. Limitations and Future Directions

The study was not free from limitations. The sample was small and not reflective of all refugee groups in Australia. Further, there were very small numbers for various visa categories, and it was not possible to compare the subgroups. A larger sample with participants from a range of communities with varying demographics and visa statuses should be recruited in the future to build a better understanding of the roles that different contexts play in the identity development among the youth from a refugee background. Further, a larger group would allow the researchers to draw conclusions surrounding the similarities and differences among former refugee youth living in different circumstances.

Further, data were collected in a single session. Despite efforts to enhance rapport, it is possible that some participants may have been inhibited from sharing their experiences. Future studies may use multiple meetings with the young people to understand the role of their social environment in the development of their identity. Further, the present study did not provide the participants with an opportunity to check their responses. Future studies could engage in member checking [42] (p. 277), which would provide the young people with an opportunity to provide feedback on how well they felt the interpretations captured their experience [44]. This exploration holds the potential to provide deeper insights into the nuanced nature of identity formation within this population.

### 4.3. Conclusions

This study highlights the salient identity challenges faced by young people from refugee backgrounds residing in Australia. Intricately woven throughout the dataset is the persistent endeavour of these young people to derive meaning from their past and present life events within the context of their family, school, and community setting, striving to reconstruct a coherent sense of self. These observations align with a conceptualisation of identity that emphasises ongoing processes, demanding the meaningful integration of each new experience and relationship into an individual’s evolving self-structure.

Overall, the study highlights the imperative for interventions aimed at supporting young people from refugee backgrounds that foster the exploration and interpretation of life experiences and facilitate the development of a more cohesive sense of identity. This approach is particularly salient in working with young individuals who have encountered significant adversity, displacement, and relocation. The significance of school connectedness is notably emphasised, with a sense of belonging serving as a crucial buffer against further ruptures in personal coherence. This finding is congruent with prior research that has alluded to the pivotal role of peers in the identity construction process for refugee youth [45,46,47].

## Figures and Tables

**Figure 1 ijerph-21-00281-f001:**
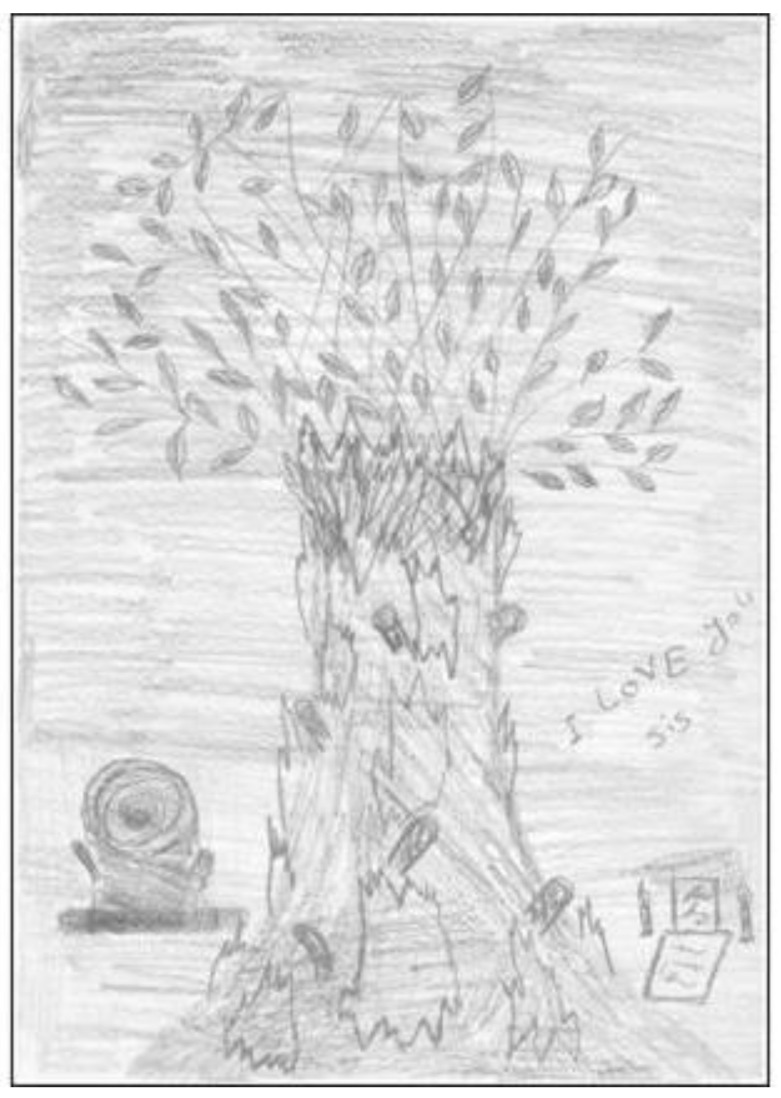
Salim’s drawing of the tree.

**Figure 2 ijerph-21-00281-f002:**
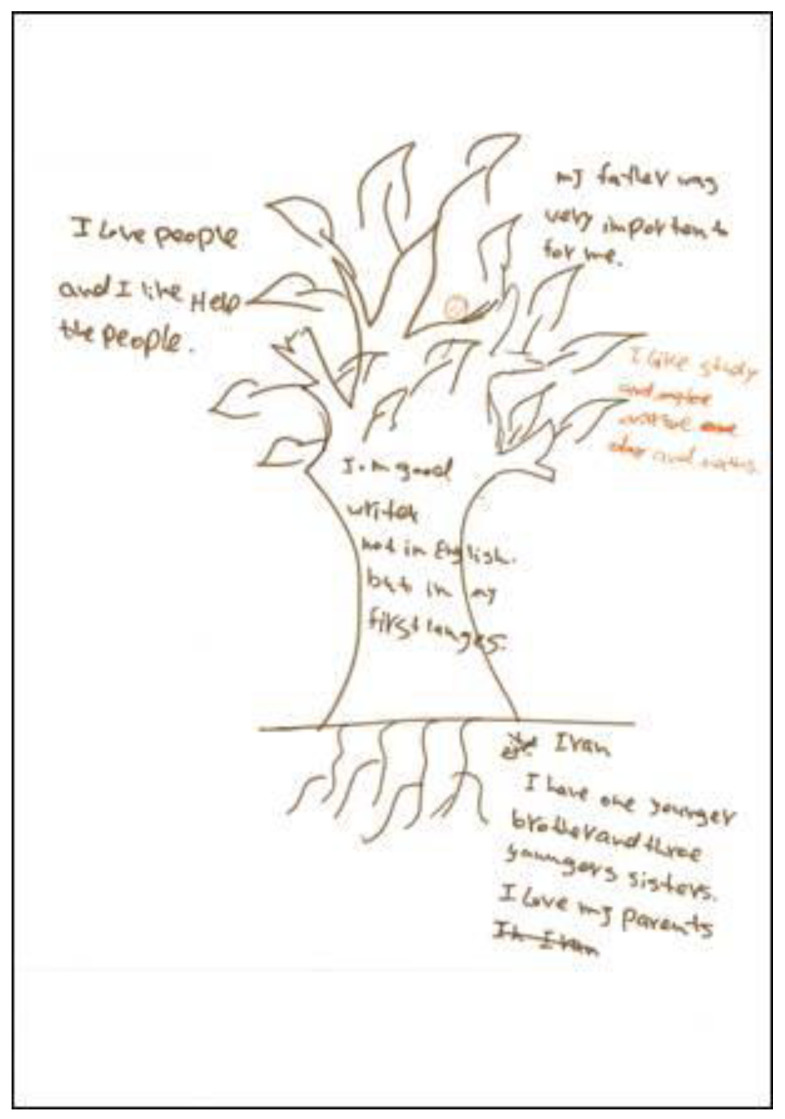
Mairi’s drawing of the tree.

**Table 1 ijerph-21-00281-t001:** Participant gender, age, ethnicity, visa status, and length of time in Australia.

Participants ^a^		Gender	Age	Ethnicity	Visa Status	Time in Australia
1	Salim	Male	18	Iraqi	Bridging	11 months
2	Khin	Male	18	Burmese	Bridging	15 months
3	Mairi	Female	17	Iranian	Permanent	11 months
4	Lucy	Female	17	Sierra Leonean	Provisional	6 months
5	Juliet	Female	15	Liberian	Provisional	11 months
6	Farhia	Female	16	Somali	Permanent	15 months
7	Ariana	Female	16	Northern African	Permanent	11 months
8	Matida	Female	16	South Sudanese	Permanent	1 month
9	Matthew	Male	18	South Sudanese	Permanent	1 month
10	Sana	Female	15	Pakistani	Permanent	15 months
11	Mirembe	Female	16	Ugandan	Permanent	4 months
12	Rima	Female	18	Syrian	Permanent	7 months
13	Ajmal	Male	18	Afghan	Bridging	24 months
14	Hani	Female	17	Somali	Permanent	20 months
15	Kalan	Male	15	Afghan	Permanent	13 months
16	Estelle	Female	16	Ivory Coast	Permanent	7 months
17	Alima	Female	17	South Sudanese	Permanent	8 months
18	Mariam	Female	16	Afghan	Permanent	19 months
19	Nazir	Male	15	Afghan	Permanent	15 months

^a^ Pseudonyms issued to each participant to preserve anonymity. A Bridging Visa is a temporary visa granted to individuals who are in Australia and have lodged an application for a substantive visa (such as a protection visa or partner visa) but have not yet received a decision on that application. A Permanent Visa is a visa that allows an individual to live and work in Australia indefinitely. The most common permanent visa is the Protection Visa (subclass 866). The Provisional Visa category is for individuals who are in a genuine and ongoing relationship with an Australian citizen, permanent resident, or eligible New Zealand citizen. It allows the visa holder to enter and stay in Australia temporarily while their permanent partner’s visa application (subclass 100) is being processed.

## Data Availability

Data will be made available on request.

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
