# Peer review of "A Qualitative Study of Adolescents from Refugee Backgrounds Living in Australia: Identity and Resettlement"

_ijerph, 2024, doi:10.3390/ijerph21030281_

Round 1

Reviewer 1 Report

Comments and Suggestions for Authors

Thank you for the opportunity to review this manuscript. Study results are an important contribution to the refugee resettlement literature, particularly with regard to identity and belonging.

Prior to publication, please address the following questions and comments, designed to strengthen your manuscript.

1. In the abstract, the authors use the term "assimilating" with regard to identities and experiences. This is the only place that this term is used. In resettlement literature, assimilation refers to giving up one's identity from one's country and culture of origin and fully taking on the culture of the host country. It seems that "integration" may be a more appropriate term here.

2. Page 3, last sentence of paragraph top of page: Are there words missing or/and duplicate words? Please check sentence for meaning (line 98-100).

3. Page 3, Table 1. Please provide a description of the following immigration statuses: bridging, permanent, provisional. Permanent is self-explanatory, but the other two need clarification. One could assume that one's status, that is, whether or not one will be able to stay in Australia, will certainly contribute to sense of identity and belonging.

Page 5, line 205: the researcher was or the researchers were. This brings up an issue however that needs clarification. It's not clear why one of the authors (last author) is specifically referenced here as the interviewer. It also seems that the last author was fully responsible for the data collection and for the decision of saturation. It's not clear why this information is provided. It begs the question of why the last author is in fact last. It's distracting.

Page 6, line 218: This comment relates the previous comment. This needs to be clarified. Why was all data analysis completed by only one author, why is she the last author...again, providing this information begs questions.

Page 6, Findings: There are several places where "many" is used. Given that there are only 19 participants, this needs greater clarification as "many" gives no real indication of how many participants felt a certain way (e.g., line 251, 267, 282). 

Page 6, line 267: This is why migration status makes a difference. Salim had a bridging visa. What is that? How does it differ from a provisional visa? Do you have other quotes that you can include from participants who had permanent visas? Provisional? There needs to be an analysis of the ramifications of each visa category.

Page 7, Section 5.4: Can you add another quote for support? That would strengthen discussion of theme. 

Overall in Findings, it would strengthen analysis and conclusions if there were at least two quotes for each theme discussed as well as an indication of how many felt/experiened similar things, etc. Otherwise, one theme can seem as though it's based on one participant's experience. For example, Section 5.6 is well supported.

Findings: Could you include one of the participant's Tree of Life drawings, de-identified? Without it, or at least a template/example, it is difficult for the reader to integrate how this was helpful for the interviewer in eliciting the narratives.

Page 9, line 392: is deceased or has died

Page 10, line 425: missing period after beneficial. 

Conclusion should include practical recommendations for those working with refugee youth in school and community settings. The authors note that study findings support development of interventions "foster the exploration and interpretation of life experiences, and facilitate the development of a more cohesive sense of identity". This should be linked more closely to current literature.

Author Response

Reviewer 1

  1. Thank you for the opportunity to review this manuscript. Study results are an important contribution to the refugee resettlement literature, particularly with regard to identity and belonging.

Prior to publication, please address the following questions and comments, designed to strengthen your manuscript.

We wish to thank the reviewer and express our appreciation for their thoughtful comments.

  1. In the abstract, the authors use the term "assimilating" with regard to identities and experiences. This is the only place that this term is used. In resettlement literature, assimilation refers to giving up one's identity from one's country and culture of origin and fully taking on the culture of the host country. It seems that "integration" may be a more appropriate term here.

We thank the reviewer, and our use of “assimilating” in this context was clearly an error.  We have now changed the term as suggested, to “integration”. Se Pg 3.

  1. Page 3, last sentence of paragraph top of page: Are there words missing or/and duplicate words? Please check sentence for meaning (line 98-100).

We have now revised this section to read more clearly.

  1. Page 3, Table 1. Please provide a description of the following immigration statuses: bridging, permanent, provisional. Permanent is self-explanatory, but the other two need clarification. One could assume that one's status, that is, whether or not one will be able to stay in Australia, will certainly contribute to sense of identity and belonging.

We thank the reviewer for reminding us that these terms need to be defined for an international audience.  We have now added an explanation to Table 1, which reads:

A Bridging Visa is a temporary visa granted to individuals who are in Australia and have lodged an application for a substantive visa (such as a protection visa or partner visa) but have not yet received a decision on that application.

A Permanent Visa is a visa that allows an individual to live and work in Australia indefinitely. The most common permanent visa is the Protection Visa (subclass 866).

The Provisional visa category is for individuals who are in a relationship with an Australian citizen, permanent resident, or eligible New Zealand citizen. It allows the visa holder to enter and stay in Australia temporarily while their permanent partner visa application (subclass 100) is being processed.

  1. Page 5, line 205: the researcher was or the researchers were. This brings up an issue however that needs clarification. It's not clear why one of the authors (last author) is specifically referenced here as the interviewer. It also seems that the last author was fully responsible for the data collection and for the decision of saturation. It's not clear why this information is provided. It begs the question of why the last author is in fact last. It's distracting.

This error on our part has now been corrected. The last named author was responsible for the collection of data, in a project conceived by the first two authors.

  1. Page 6, line 218: This comment relates the previous comment. This needs to be clarified. Why was all data analysis completed by only one author, why is she the last author...again, providing this information begs questions.

We understand the concerns of the reviewer and have redrafted the section to better reflect the collaborative nature of the explication of the data.

  1. Page 6, Findings: There are several places where "many" is used. Given that there are only 19 participants, this needs greater clarification as "many" gives no real indication of how many participants felt a certain way (e.g., line 251, 267, 282). 

We have replace “many’ by clearer phrases, such “all of them” or half of them.

Page 6, line 267: This is why migration status makes a difference. Salim had a bridging visa. What is that? How does it differ from a provisional visa? Do you have other quotes that you can include from participants who had permanent visas? Provisional? There needs to be an analysis of the ramifications of each visa category.

Ideally, it would have been good to undertake a sub analysis based upon Visa category, but as some Visa categories were so small, e.g., bridging equals 3, we focussed upon the commonality of all participants coming from refugee backgrounds.

  1. Page 7, Section 5.4: Can you add another quote for support? That would strengthen discussion of theme. 

We have added a couple of more quotes.

  1. Overall in Findings, it would strengthen analysis and conclusions if there were at least two quotes for each theme discussed as well as an indication of how many felt/experiened similar things, etc. Otherwise, one theme can seem as though it's based on one participant's experience. For example, Section 5.6 is well supported.

         Quotes are added.

  1. Findings: Could you include one of the participant's Tree of Life drawings, de-identified? Without it, or at least a template/example, it is difficult for the reader to integrate how this was helpful for the interviewer in eliciting the narratives.

Two drawings are now added and integrated in the findings.

  1. Page 9, line 392: is deceased or has died

We have changed to “is deceased”.

  1. Page 10, line 425: missing period after beneficial. 

This has now been corrected.

  1. Conclusion should include practical recommendations for those working with refugee youth in school and community settings. The authors note that study findings support development of interventions

We appreciate this suggestion and have added a new section in the conclusion section with a number of recommendations, derived from our findings.

Reviewer 2 Report

Comments and Suggestions for Authors

Thank you for the opportunity to review this manuscript focusing on an interesting and challenging phenomena – the complexity of identity formation of adolescents from refugee backgrounds within Australia. Please see my comments below – I hope they are useful.

Methods section:

The use of the Tree of Life is an appropriate methodological approach when working with young people, enabling narrative exploration into identity formation.

Recruitment section: “Recruitment was undertaken with sensitivity to the need to build rapport with potential participants prior to the commencement of research.” – what does this mean exactly i.e. how was this done (e.g. methods/time frame of rapport building, recruitment communications with participants etc). There are potential implications for data collection and analysis.

Data Collection: I appreciate that the instructions as noted appear in conversational terms. For the reader, can the instructions be contained to dot points, for clarity of process?

Researcher positioning – who are the researchers? e.g. people from refugee background, experts in refugee identity research? Please provide some context here, this will provide much needed perspective on the analysis etc, but also to give an idea of roles in research – who was lead researcher, who did interviews, analysis etc?. I can see this is touched on in Data Explication section, but could be more upfront for the reader.

Data explication: Braun & Clarke’s Thematic Analysis is better described as theoretically flexible, not atheoretical (see the TA FAQ).

Related to this, I find the headings in the introduction unclear. Part 1.1 covers identity from the sociocultural approach with some depth, then appears to segue to the Self approach. However, heading 1.2 then suggests that Identity from the self-psychological perspective will be discussed at length, and this isn’t the case – it instead speaks to the apparent contradiction between the two approaches. I think some amendment to how this section is outlined is necessary.

The above two points are important and interconnected - I as the reader have assumed the data analysis is based on the theoretical framing (and this potential contradiction noted) provided in the introduction.

The inclusion criteria (and most importantly the rationale behind the criteria i.e. analysis and interpretability etc) directly contradict involvement of 1 of the participants (the participant requiring a Farsi interpreter). Why was the inclusion of this individual necessary?

Findings:

The themes are mostly accurately depicted and detailed, with relevant quotes to exemplify the theme.

However, theme 2 “Experience of belonging” is less cohesive. The description makes sense mostly (the first two paragraphs, but no quote), whereas the third paragraph has the lone quote provided, and while I understand the purpose of it (i.e. a counterpoint to the sense of belonging found) it does not really reflect/support the theme well. This may simply be the way it has been written, or the lack of quotes in this theme in general.

Theme 3 “experience of bonds with loved ones” is quite well described and important, but the lack of quotes makes it harder to connect the participants experience to the description.

Theme 5 is for me the strongest theme of the five. The description itself covers the notion of meaning making, and the factors that might contribute to it. I like the way the authors have linked this theme to some of the concept’s in Theme 3.

Discussion:

First up, I think its worth making more clear the original contribution that this paper makes. Has this been studied before etc?

P. 8, line 351 states “From the perspective of theory, the roles explicated in the paper, addressing experiences of changes in family roles, experiences of belonging, experiences of bonds with lost loved ones, the process of dealing with emotions in a new context, and, experience of self in the context of change, all contribute to the formation of a person's identity, in terms of their membership of social groups [28].”. Further, on page 9, line 407, Social ID Theory is introduced. This appears to be important and essentially the theoretical framing of the paper. If this is the case, this should be more explicitly outlined from the introduction to strongly position the study. If this is not the case, the introduction needs to more clearly outline the literature of identity and how it frames the research theoretically,  to better support the weaving together of these theoretical strands into a coherent analysis and discussion. The conceptualisation of identity in this paper is clearly multifaceted – not a bad thing, from my perspective – but needs the aforementioned work to make it cohesive for the reader. Regarding findings, these are discussed well, in terms of literature that supports them. However, were there things that contradicted the literature, or were unclear i.e. alternative explanations. Again, I think clearer articulation of the theoretical framing would make these last points more apparent to the author.

For clarity of reading, I think it worth also adding a subheading in Discussion section to denote Limitations & Strengths. At present, the Limitations and strengths somewhat reads as an expanded version of content that should have been in the Methods section. What makes this paper a strong one, worthy of publication? What are the limitations the reader should take into account when interpreting the findings etc? I would argue the second paragraph doesn’t belong in the Conclusion. It should be in the Discussion, as (for example) Implications for Practice, as some of it speaks to the ‘how’ of working with this population to achieve certain goals.

Similarly, directions for future research (e.g. Line 470 on p. 10) should be in the Discussion section also, not the Conclusion. There may be others also.

Author Response

Reviewer 2

  1. Thank you for the opportunity to review this manuscript focusing on an interesting and challenging phenomena – the complexity of identity formation of adolescents from refugee backgrounds within Australia. Please see my comments below – I hope they are useful.

We appreciate the feedback.

  1. Methods section:

The use of the Tree of Life is an appropriate methodological approach when working with young people, enabling narrative exploration into identity formation.

  1. Recruitment section: “Recruitment was undertaken with sensitivity to the need to build rapport with potential participants prior to the commencement of research.” – what does this mean exactly i.e. how was this done (e.g. methods/time frame of rapport building, recruitment communications with participants etc). There are potential implications for data collection and analysis.

We have now revised this section and elaborated upon the process.  The section now reads:

Recruitment was undertaken over an extended period with sensitivity to the need to build rapport with potential participants prior to the commencement of research. That is, members of the research team had ongoing contact with the school for about five years and attended school functions and developed familiarity with members of the school community. This process was deemed essential in developing trust within the school community.

  1. Data Collection: I appreciate that the instructions as noted appear in conversational terms. For the reader, can the instructions be contained to dot points, for clarity of process?

The process has been reformatted in accordance with the suggestion.

  1. Researcher positioning – who are the researchers? e.g. people from refugee background, experts in refugee identity research? Please provide some context here, this will provide much needed perspective on the analysis etc, but also to give an idea of roles in research – who was lead researcher, who did interviews, analysis etc?. I can see this is touched on in Data Explication section, but could be more upfront for the reader.

We have addressed this comment by including a new section at the end of the Method section, referred to as: Reflexive comment.

  1. Data explication: Braun & Clarke’s Thematic Analysis is better described as theoretically flexible, not atheoretical (see the TA FAQ).

Thank you, we have made this change.

  1. Related to this, I find the headings in the introduction unclear. Part 1.1 covers identity from the sociocultural approach with some depth, then appears to segue to the Self approach. However, heading 1.2 then suggests that Identity from the self-psychological perspective will be discussed at length, and this isn’t the case – it instead speaks to the apparent contradiction between the two approaches. I think some amendment to how this section is outlined is necessary.

We agree with the reviewers. In an earlier draft of the paper, we included theory drawn from self-psychology, but have now revised the paper to be consistent, and not include this framework, which was confusing. We have revised this section accordingly. The paper now uses Social Identity theory and Sociocultural perspcetve.

  1. The above two points are important and interconnected - I as the reader have assumed the data analysis is based on the theoretical framing (and this potential contradiction noted) provided in the introduction.

In the explication of the data, we adopted an inductive approach, to be open to themes which emerged from the data. However, it is not possible to not have a theory which informed our thinking and this is now better described within the revised introduction 

The inclusion criteria (and most importantly the rationale behind the criteria i.e. analysis and interpretability etc) directly contradict involvement of 1 of the participants (the participant requiring a Farsi interpreter). Why was the inclusion of this individual necessary?

Interpreters were offered to all participants , however, this offer was accepted by only one participant.

  1. Findings:

The themes are mostly accurately depicted and detailed, with relevant quotes to exemplify the theme.

However, theme 2 “Experience of belonging” is less cohesive. The description makes sense mostly (the first two paragraphs, but no quote), whereas the third paragraph has the lone quote provided, and while I understand the purpose of it (i.e. a counterpoint to the sense of belonging found) it does not really reflect/support the theme well. This may simply be the way it has been written, or the lack of quotes in this theme in general.

Quotes and explanations are added.

Theme 3 “experience of bonds with loved ones” is quite well described and important, but the lack of quotes makes it harder to connect the participants experience to the description.

Quote is added.

  1. Theme 5 is for me the strongest theme of the five. The description itself covers the notion of meaning making, and the factors that might contribute to it. I like the way the authors have linked this theme to some of the concept’s in Theme 3.

Thank you, we appreciate the feedback.

  1. Discussion:

First up, I think its worth making more clear the original contribution that this paper makes. Has this been studied before etc?

We have revised the beginning of this section to address the concern raised.

This appears to be important and essentially the theoretical framing of the paper. If this is the case, this should be more explicitly outlined from the introduction to strongly position the study. If this is not the case, the introduction needs to more clearly outline the literature of identity and how it frames the research theoretically, to better support the weaving together of these theoretical strands into a coherent analysis and discussion. The conceptualisation of identity in this paper is clearly multifaceted – not a bad thing, from my perspective – but needs the aforementioned work to make it cohesive for the reader.

The introduction is revised extensively. Social identity theory and Sociocultural perspective is now added to the paper  (Introduction) and incorporated in the Discussion.

  1. Regarding findings, these are discussed well, in terms of literature that supports them. However, were there things that contradicted the literature, or were unclear i.e. alternative explanations. Again, I think clearer articulation of the theoretical framing would make these last points more apparent to the author.

        We have tried to provide a clearer theoretical framework (See Introduction)

  1. For clarity of reading, I think it worth also adding a subheading in Discussion section to denote Limitations & Strengths. At present, the Limitations and strengths somewhat reads as an expanded version of content that should have been in the Methods section. What makes this paper a strong one, worthy of publication? What are the limitations the reader should take into account when interpreting the findings etc?

Headings are added.

  1. I would argue the second paragraph doesn’t belong in the Conclusion. It should be in the Discussion, as (for example) Implications for Practice, as some of it speaks to the ‘how’ of working with this population to achieve certain goals.

Corrected

  1. Similarly, directions for future research (e.g. Line 470 on p. 10) should be in the Discussion section also, not the Conclusion. There may be others also.

Corrected

We wish to thank the reviewers for their thoughtful and helpful feedback.  We believe that we have not addressed each of the points raised by the reviewers.  We look forward to your response.

With best wishes

Nigar G Khawaja

Professor